# Ion-Exchange-Induced Transformation and Mechanism of Cooperative Crystal Chemical Adaptation in Sitinakite: Theoretical and Experimental Study

**Taras L. Panikorovskii [1,2,\*], Galina O. Kalashnikova [3], Anatoly I. Nikolaev [3], Igor A. Perovskiy [1,4], Ayya V. Bazai [5], Victor N. Yakovenchuk [5], Vladimir N. Bocharov [6], Natalya A. Kabanova [1,7] and Sergey V. Krivovichev [2,3]**

[1] Laboratory of Nature-Inspired Technologies and Environmental Safety of the Arctic, Kola Science Centre, Russian Academy of Sciences, 14 Fersman Street, 184200 Apatity, Russia; igor-perovskij@yandex.ru (I.A.P.); weterrster@gmail.com (N.A.K.)

[2] Department of Crystallography, Institute of Earth Sciences, Saint–Petersburg State University, University Emb. 7/9, 199034 St. Petersburg, Russia; s.krivovichev@ksc.ru

[3] Nanomaterials Research Centre of Kola Science Centre, Russian Academy of Sciences, 14 Fersman Street, 184209 Apatity, Russia; g.kalashnikova@ksc.ru (G.O.K.); a.nikolaev@ksc.ru (A.I.N.)

[4] Institute of Geology, Komi Science Centre, Ural Branch, Russian Academy of Sciences, 167982 Syktyvkar, Russia

[5] Geological Institute of Kola Science Centre, Russian Academy of Sciences, 14 Fersman Street, 184209 Apatity, Russia; bazai@geoksc.apatity.ru (A.V.B.); v.yakovenchuk@ksc.ru (V.N.Y.)

[6] Geo Environmental Centre "Geomodel", Saint–Petersburg State University, Ul'yanovskaya Str. 1, 198504 St. Petersburg, Russia; regbvn@gmail.com

[7] Samara Center for Theoretical Materials Science, Samara State Technical University, Molodogvardeyskaya Str. 244, 443100 Samara, Russia

\* Correspondence: t.panikorovskii@ksc.ru or taras.panikorovsky@spbu.ru; Tel.: +7-81555-79628

**Abstract:** The microporous titanosilicate sitinakite, $KNa_2Ti_4(SiO_4)_2O_5(OH)\cdot4H_2O$, was first discovered in the Khibiny alkaline massif. This material is also known as IONSIV IE-911 and is considered as one of the most effective sorbents for $Cs^+$ and $Sr^{2+}$ from water solutions. We investigate a mechanism of cooperative crystal chemical adaptation caused by the incorporation of $La^{3+}$ ions into sitinakite structure by the combination of theoretical (geometrical–topological analysis, Voronoi migration map calculation, structural complexity calculation) and empirical methods (PXRD, SCXRD, Raman spectroscopy, scanning electron microscopy). The natural crystals of sitinakite ($a = 7.8159(2)$, $c = 12.0167(3)$ Å) were kept in a 1M solution of $La(NO_3)_3$ for 24 h. The ordering of $La^{3+}$ cations in the channels of the ion-exchanged form $La^{3+}Ti_4(SiO_4)_2O_5(OH)\cdot4H_2O$ ($a = 11.0339(10)$, $b = 11.0598(8)$, $c = 11.8430(7)$ Å), results in the symmetry breaking according to the group–subgroup relation $P4_2/mcm \rightarrow Cmmm$.

**Keywords:** sitinakite; titanosilicate; IONSIV IE-911; $La^{3+}$; ion-exchange; crystal structure; ordering; titanosilicate; ion migration; Arctic

## 1. Introduction

The peaceful use of atomic energy is accompanied by the formation of significant quantities of radioactive waste [1]. The formation of liquid radioactive waste (LRW) is primarily associated with the reprocessing of spent nuclear fuel. The LRW contains radioactive isotopes $^{60}Co$, $^{89,90}Sr$, $^{103,106}Ru$, $^{134,137}Cs$, $^{226}Ra$, $^{234,238}U$, $^{238,239}Pu$, representing a permanent environmental hazard and therefore requiring decontamination. To date, the most promising and reliable method of LRW immobilization is the long-term fixation of radionuclides in natural-like mineral matrices with their transformation into the SYN-ROC (SYNthetic ROCk) ceramics resistant to chemical decomposition and self-exposure developed by A. E. Ringwood at the Australian National University [2–8].

During recent years, special attention has been paid to natural titanosilicates, associated mainly with alkaline rocks of the Kola Peninsula, Russian Federation [9]. There are 69 titanosilicate minerals found in the Khibiny and 78 in Lovozero alkaline massifs [10]. Among them, there are many microporous compounds, of which synthetic analogs are considered as sorbents, molecular sieves and ion-exchangers [11–15]. The most remarkable examples are zorite (ETS-4), ivanyukite (GTS, SIV), kamenevite (AM-2, STS) and sitinakite (IONSIV-911, TAM-5, CST) [16] that were discovered for the first time in the Khibiny alkaline massif [17]. In recent studies, particular focus in catalysis and ion-exchange concentrated on the framework and layered titanosilicates: lovozerite, hilairite, lintisite, kukisvumite, murmanite and labuntsovite-type compounds [18–23]. Today the ETS-4 (zorite), IONSIV-911 (sitinakite) and SIV (ivanyukite) are considered as the absorbents of the $^{90}$Sr and $^{137}$Cs radioactive isotopes with subsequent conversion of high-level to low-level radioactive waste [16,24].

Sitinakite, $Na_2KTi_4Si_2O_{13}(OH)\cdot4H_2O$, was found in 1989 at the natrolite-vinogradovite-aegirine vein in melteigite-urtite at Mt. Kukisvumchorr and named for its chemical composition (Si-Ti-Na-K-ite). At this locality, it forms brownish-pink short-prismatic (pseudocubic) crystals up to 2 mm long and 1 mm in diameter in natrolite nodules enriched with vinogradovite [25]. The crystal forms are the {100} and {110} tetragonal prisms and the pinacoid {001}. The crystals are usually mosaic, with some characteristic cross-like aggregates, and encrust the walls of voids in aegirine within the core of the vein. Other associated minerals are lobanovite, lorenzenite, shcherbakovite, pectolite, lamprophyllite, djerfisherite, ancylite-(Ce). Sitinakite is a low-temperature hydrothermal mineral formed as a result of recrystallization of primary shcherbakovite, nenadkevichite and lomonosovite [25].

The synthetic analog of sitinakite was obtained in 1993 [24,26,27] and was used as an absorbent for Cs, Sr and Rb radionuclides in different solutions with a wide pH range [28,29]. The substitution of $Nb^{5+}$ for $Ti^{4+}$ greatly enhances the material's selectivity for $Cs^+$ as revealed in a patent held by Sandia National Laboratories [30]. There are two forms of sitinakite analogs available as a commercial product by Union Oil Products Inc. (Chicago, IL, USA): IONSIV IE-910 and niobium-containing IONSIV IE-911. According to the Sandia National Laboratories data, applying IONSIVE-911 to the Hanford cleanup alone will result in savings of more than USD 300 million over baseline technologies [31].

This material and its modifications, also known as CST, PCST, IONSIV R9120, etc., are well-known efficient exchangers for radionuclides. Their performance has already been reported after their utilization for nuclear waste effluent treatment processes (Fukushima Daiichi plant after the events from 11 March 2011) [32].

It has been found that the material's performance is improved if heated to 100 °C and these temperature conditions are close to those used for the removal of radioactive Cs from waste streams [13]. It is important to note that the use of sorbents, which are not capable of displaying their ion-exchange properties under normal conditions, leads to the significant complication of working conditions with radioactive waste. The high-temperature study of dehydration dynamics of sitinakite analog was reported in [33]. The structural transitions in synthetic sitinakite have been reported for the H- and Sr-substituted forms based on the Rietveld refinement [34,35].

In general, LRW includes waste generated as a result of the nuclear fuel cycle of a nuclear power plant. Nonetheless, this kind of waste can be obtained after the use of radioisotopes in medical diagnostic research, pharmaceutical and biotechnological developments. Such wastes usually include organic substances and metal compounds, some of which may also include lanthanum ($^{140}$La) isotopes [36,37].

In our recent works, we demonstrated different mechanisms of structure adaptation induced by the cation ordering causing symmetry breaking in vesuvianite [38,39], garnet- [40] and eudialyte- [41] group minerals. Mechanisms of incorporation of mono and divalent cations are intensively reported during last years [13,42], however, there is a lack of information about the incorporation of trivalent cations in sitinakite structure.

Owing to the endemic status of the mineral the single-crystal X-ray diffraction (SC XRD), studying ion-exchange mechanisms in sitinakite is quite difficult for studies. Additionally, the size of synthetic counterparts often is not appropriate for SC XRD. Herein we report the incorporation of $La^{3+}$ into natural sitinakite at 200 °C and the structural adaptation mechanism based on the SC XRD data. The dynamics of incorporation of $La^{3+}$ into synthetic sitinakite was studied for the powder material exposed for 1, 4, 12 and 24 h at 200 °C in 1M $La(NO_3)_3$ solution. The theoretical analysis of topology and the cation migration paths was performed for both sitinakite and its La-exchanged form using a geometrical and topological approach [43,44].

## 2. Materials and Methods

### 2.1. Sample

For the structural studies, sitinakite crystals were selected from the nepheline-sodalite-microcline-aegirine vein no. 8 (according to [45]) at Mt. Koashva, Kola Peninsula, Russia (Figure 1a). The concentric vein of about 5 m in diameter situated in the quarry wall of the Koashva apatite-nepheline mine (Figure 1b). The marginal zone (0.2−1 m) is composed of monomineralic light-green microcline aggregates. An intermediate zone (0.5−1.5 m) consists of up to 50 cm dark red sodalite aggregates with grey nepheline and irregularly shaped pectolite aggregates up to 15 cm in size with brown grains of semi-transparent titanite. The core zone (3 m) contains radiating fibrous aegirine up to 50 cm in diameter with radiating aggregates of lamprophyllite, irregularly shaped nodules of pectolite and natrolite up to 80 cm in size. Pectolite and natrolite aggregates contain voids encrusted by rare phosphate and silicate minerals: arctite, belovite-(Ce), carbonate-fluorapatite, catapleite, shcherbakovite. Sitinakite was found in association with sazykinaite-(Y), rhabdophane-(Ce), aegirine, ivanyukite-group minerals. Sitinakite forms pale pink short-prismatic (up to 1 mm) crystals and also occurs as small colorless or brown crystals containing inclusions of lemmleinite-K and colored by solid organics. The crystals are usually mosaic, with some characteristic cross-like aggregates, and encrust the walls of voids in aegirine within the core of the vein.

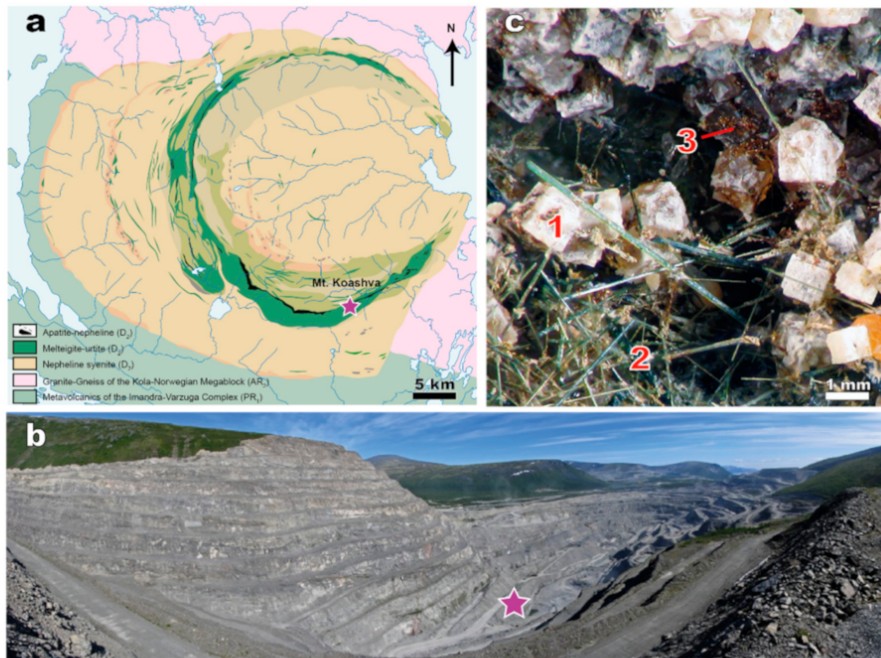

**Figure 1.** (**a**) Geology of the Khibiny massif and Mt. Koashva within purple star after [46] with modifications; (**b**) the Koashva quarry, Khibiny alkaline massif, Kola Peninsula, Russia. The purple star indicates the location for the sitinakite bearing vein no. 8; (**c**) short-prismatic sitinakite (1) crystals with aegirine needles (2) and amorphous bitumens (3).

## 2.2. Synthesis

A hydrated precipitate obtained by the original fluorine-ammonium method for leucoxene concentrate (Yarega deposit, Komi Republic, Russia) processing [47] was used as a precursor for the synthesis of titanosilicates. The enrichment of the leucoxene concentrate was carried out in a tubular furnace equipped with a gas outlet system that allows volatile products (gaseous ammonia, water, volatile fluoride compounds) to be pumped out into a vessel with water. The concentrate was mixed with ammonium hydrofluoride in a glassy carbon crucible without additional abrasion and placed in the center of the furnace, where two step heating was carried out: at 220 (heating rate was 10 °C/min) and 300 °C (heating rate was 2 °C/min). The sample was aged for 30 min at each temperature. After fluorination, water leaching (deionized water heated to 70 °C with a volume of 50 cm$^3$, pH = 5.20 ± 0.10) was carried out, which made it possible to transfer undecomposed fluoride complexes into solution via hydrolysis. The remainder of the titanium enriched concentrate after leaching was separated from the mother solution by filtration and dried at 103 °C. The fluorinated product is a high-titanium concentrate containing more than 85% TiO$_2$ and represented by a mixture of rutile and anatase. The fluorinated product can be processed into pigmented titanium dioxide or into metallic titanium.

The waste product on fluorination, the mother solution, is a multi-component system of dissolved fluorammonium salts. The addition of ammonia (6% NH$_4$OH) to the mother solution leads to gradual aggregation of colloidal particles and formation of a hydrated precipitate. The chemical composition of the hydrated precipitate was determined using X-ray fluorescence analysis: 50.5%—TiO$_2$; 45.5%—SiO$_2$; 2.9%—Fe$_2$O$_3$; 0.4%—Al$_2$O$_3$; 0.2%—CaO; 0.2%—K$_2$O; 0.2%—NbO and ZrO$_2$. The dried hydrated precipitate in the amount of 0.5 g was treated with 37 mL of 1 M NaOH solution and dispersed for 20 min with a magnetic stirrer. The final mole ratio of Na$_2$O:TiO$_2$:SiO$_2$:H$_2$O in the resulting alkali titanium-silicon mixture was 6:1:1.2:657.7. The obtained mixture was transferred to a Teflon-lined autoclave (45 mL, filling degree 80%) and kept at 250 °C for 12 h (pressure in the autoclave can be estimated as ~80 atm). After cooling down to room temperature (20 °C), the product was collected by centrifugation and washed with distilled water (450 mL) until pH 5.6–6. The sample was dried at 103 °C for 4 h.

## 2.3. Ion-Exchange

Several crystals of sitinakite have been kept in 1 M La(NO$_3$)$_3$ solution (10 mL) for 24 h at 200 °C without a periodic shaking in the hermetically sealed autoclaves for hydrothermal synthesis (TOPT-HT10, Toption Instrument, Zhengzhou, China). After the removal of the La(NO$_3$)$_3$ solution with a Pasteur pipette, the crystals were washed with a threefold volume of distilled water and dried in air for 2 h. After the contact with the La(NO$_3$)$_3$ solution, the crystals of sitinakite changed their color from pale pink to white-matte. The morphology of the crystals remained unchanged. Synthetic sitinakite was stored under the same conditions for 1, 4, 12 and 24 h.

## 2.4. Composition

The morphology of La-exchanged sitinakite samples was studied using scanning electron microscope LEO-1450 (Carl Zeiss Microscopy, Oberkochen, Germany) equipped with Oxford Instruments Ultim Max 100 analyzer (Geological Institute, Kola Science Centre, Russian Academy of Sciences, Apatity, Russia). The composition of initial synthetic and La-exchanged samples was measured by electron microscope Vega 3 LMH (Tescan) equipped with Oxford Instruments X-ACT energy dispersive analyzer.

The chemical composition of natural sitinakite was determined by wavelength-dispersive spectrometry on a Cameca MS-46 electron microprobe (Geological Institute, Kola Science Centre, Russian Academy of Sciences, Apatity, Russia) operating at 20 kV and 20–30 nA, with a 20 μm beam diameter. The standards used were: lorenzenite (Na, Ti), pyrope (Al), wollastonite (Si, Ca), wadeite (K), synthetic MnCO$_3$ (Mn), hematite (Fe), metallic copper (Cu) and synthetic LiNbO$_3$ (Nb). Analyses were performed with the probe defocused up to

20 μm, and by continuous movement of the sample to minimize sample damage and the Na, K, and $H_2O$ loss for 10 s counting time.

### 2.5. Raman Spectroscopy

The Raman spectra (RS) of sitinakite and La-exchanged form collected from uncoated individual grains were recorded with a Horiba Jobin-Yvon LabRAM HR800 spectrometer equipped with an Olympus BX-41 microscope in backscattering geometry (Saint-Peterburg State University). Raman spectra were excited by a solid-state laser (532 nm) with actual power of 2 mW under the 50x objective (NA 0.75). The spectra were obtained in the range of 70–4000 $cm^{-1}$ at the resolution of 2 $cm^{-1}$ at room temperature. To improve the signal-to-noise ratio, the number of acquisitions was set to 15. The spectra were processed using the algorithms implemented in Labspec and OriginPro 8.1 software packages (Originlab Corporation, Northampton, MA, USA).

### 2.6. Powder and Single-Crystal X-Ray Diffraction

Theoretical powder diffraction patterns for sitinakite and its La-exchanged form were calculated on the basis of our crystal-structure data by means of the VESTA 3 program [48] for ($\lambda_1$ = 1.54059 and $\lambda_2$ = 1.54432 Å) using RIETAN-FP algorithms [49]. The PXRD pattern for initial sitinakite simulation based on $P4_2/mcm$ space group with unit cell parameters of $a$ = 7.8159 and $c$ = 12.0167 Å. For the simulation of La-exchanged form were used *Cmmm* space group with $a$ = 11.0339, $b$ = 11.0598 and $c$ = 11.8430 Å.

Synthetic sitinakite and its La-exchanged forms were characterized using powder X-ray diffraction by Shimadzu XRD-6000 diffractometer with Cu$K\alpha$ radiation in the range of reflection angles 2θ of 2–60° (Institute of Geology of Komi SC).

The crystal-structure studies of natural sitinakite and their La-exchanged form were carried out at the X-ray Diffraction Resource Centre of St. Petersburg State University by means of the Oxford diffraction Xcalibur EOS and Synergy S diffractometers equipped with the CCD and Hypix detectors using monochromatic Mo$K\alpha$ radiation (λ = 0.71069 Å) at room temperature. More than half of the diffraction sphere was collected with scanning step 1°, and exposure time 10–30 s. The data were integrated and corrected by means of the CrysAlisPro program package, which was also used to apply empirical absorption correction using spherical harmonics, implemented in the SCALE3 ABSPACK scaling algorithm [50]. The structure was refined using the SHELXL software package (64-bit version-2015, Göttingen University, Göttingen, Germany) [51]. The crystal structure was drawn using the VESTA 3 program [48]. Occupancies of the cation sites were calculated from the experimental site-scattering factors (except for the low-occupied sites) in accordance with the empirical chemical composition. Hydrogen sites could not be located.

The SCXRD data are deposited in CCDC under entries No. 2126342-2126343. The coordination numbers of Na, La and K were determined by the number of bonds within the coordination spheres with the radii of 3.0, 3.0 and 3.4 Å, respectively. Crystal data, data collection information, and refinement details are given in Table 1. Atom coordinates and isotropic parameters of atomic displacements are given in Tables S1 and S2, interatomic distances in Tables S3 and S4, and the anisotropic parameters of atomic displacements are given in Tables S5 and S6.

### 2.7. Geometrical–Topological Analysis

The cation-migration paths and the elementary blocks (cavities) in the sitinakite framework were determined using geometrical–topological analysis implemented in the ToposPro software package (build 5.4.3.0., TOPOS-expert, Samara, Russia. https://topospro.com/, accessed on 1 August 2021) [43]. Maps of the migration of $Na^+$ ions were constructed by the Voronoi method (geometrical analysis), which was shown to be efficient for various types of ionic conductors [44,52,53]. The calculation methodology is described in detail in [54]. In this work, the channel crucial value parameter is based on the size of the largest mobile cation ($K^+$). Therefore, the smallest radius of elementary channels was

taken as the sum of the Slater radii [55] of $K^+$ and $O^{2-}$ taking into account the deformation coefficient-$r_{chan\ crucial}$ = 85% ($r_K + r_O$) = 2.3 Å. $Na^+$ and $La^{3+}$ have smaller Slater radii than $K^+$, and can also move along the same channel system.

**Table 1.** Crystal data, data collection information, and refinement details for sitinakite and its La-exchanged form from Koashva quarry, Khibiny alkaline massif, Kola Peninsula, Russia.

| Parameter | Data | |
|---|---|---|
| | sitinakite | La-exchanged sitinakite |
| Temperature/K | 293(2) | 293(2) |
| Crystal system | tetragonal | orthorhombic |
| Space group | $P4_2/mcm$ | $Cmmm$ |
| $a$/Å | 7.8159(2) | 11.0339(10) |
| $b$/Å | 7.8159(2) | 11.0598(8) |
| $c$/Å | 12.0167(3) | 11.8430(7) |
| Volume/Å$^3$ | 734.08(4) | 1445.23(19) |
| Z | 2 | 4 |
| $\rho_{calc}$/g/cm$^3$ | 2.763 | 2.990 |
| $\mu$/mm$^{-1}$ | 2.602 | 4.670 |
| F(000) | 592.0 | 1230.0 |
| Crystal size/mm$^3$ | 0.15 × 0.14 × 0.14 | 0.15 × 0.14 × 0.14 |
| Radiation | Mo $K\alpha$ ($\lambda$ = 0.71073) | |
| 2Θ range for data collection/° | 6.782 to 52.976 | 6.882 to 52.97 |
| Index ranges | $-6 \leq h \leq 9, -6 \leq k \leq 9, -12 \leq l \leq 15$ | $-12 \leq h \leq 12, -13 \leq k \leq 11, -14 \leq l \leq 14$ |
| Reflections collected | 1660 | 3559 |
| Independent reflections | 444 [$R_{int}$ = 0.0291, $R_{sigma}$ = 0.0248] | 841 [$R_{int}$ = 0.0182, $R_{sigma}$ = 0.0122] |
| Data/restraints/parameters | 444/0/54 | 841/24/104 |
| Goodness-of-fit on $F^2$ | 1.185 | 1.136 |
| Final $R$ indexes [$I \geq 2\sigma(I)$] | $R_1$ = 0.0343, w$R_2$ = 0.0918 | $R_1$ = 0.0372, w$R_2$ = 0.1135 |
| Final $R$ indexes [all data] | $R_1$ = 0.0366, w$R_2$ = 0.0933 | $R_1$ = 0.0373, w$R_2$ = 0.1135 |
| Largest diff. peak/hole/e Å$^{-3}$ | 0.74/−0.62 | 1.06/−0.94 |

Topological analysis of the crystal structures of sitinakite and La-exchanged form also included the determination of the type basic net and tiling construction [56–59]. The tiling method proposed in [60] and based upon the division of space into the smallest blocks (tiles) with a physical meaning of the cavity. In contrast to the geometric approach, this partition is based on the construction of an atomic net graph [60,61].

## 3. Results

### 3.1. Composition

Table 2 provides details of the analytical results for natural and synthetic sitinakite and their La-exchanged forms. Natural sitinakite contains 0.29 atoms per formula unit (apfu) of $K^+$, 0.06 apfu of $Sr^{2+}$ and 2.47 apfu of $Na^+$ with the total charge of $A$-cations of +2.88. The La-exchanged form of natural sitinakite (Figure 2a) contains a residual 0.01 apfu $K^+$ and 0.91 apfu of $La^{3+}$ with the total charge of +2.74. The absence of Al and the presence of Fe in synthetic sitinakite (Figure 2b) is explained by the composition of the initial leucoxene concentrate precursor. The exchanged form of synthetic sitinakite contains 0.96 apfu of $La^{3+}$ and 0.17 afpu of $Na^+$ compared with 0.06 apfu $K^+$ and 2.66 apfu $Na^+$ in initial synthetic sitinakite. Both synthetic and natural sitinakite demonstrate exceptional ion-exchange properties close to the ideal substitution scheme $3Na^+ \leftrightarrow 1La^{3+} + 2\square$. The slight excess of Si (2.03–2.21 apfu) was also observed for pharmacosiderite-type structure titanosilicate GTS-1 [62]. The most significant Si excess observed for the synthetic form and could be due to the presence of amorphous silica. For the natural sample slight Si excess may be connected with possible substitution $Si^{4+} + Ti^{4+} \leftrightarrow Al^{3+} + Nb^{5+}$.

**Table 2.** Chemical composition of natural and synthetic sitinakite before and after La-exchange experiments.

| Sample | Sitinakite, nat. | Sitinakite, syn. | La-Sitinakite, syn. 12 h Exchanged Form | La-Sitinakite, nat. 24 h Exchanged Form |
|---|---|---|---|---|
| $SiO_2$ | 20.20 | 21.61 | 17.80 | 19.66 |
| $TiO_2$ | 44.70 | 46.92 | 45.36 | 40.55 |
| $Al_2O_3$ | 0.43 | | | 0.42 |
| FeO | 0.15 | 2.42 | 0.84 | 0.13 |
| $Na_2O$ | 12.17 | 13.48 | 0.77 | |
| $K_2O$ | 2.15 | 0.47 | | 0.08 |
| SrO | 0.92 | | | |
| $Nb_2O_5$ | 5.49 | | | 4.64 |
| $La_2O_3$ | | | 22.83 | 21.99 |
| $H_2O$ [1] | 12.90 | 14.28 | 11.98 | 12.18 |
| Total | 99.11 | 99.18 | 99.58 | 99.65 |
| Atoms per formula unit normalized on the basis of 6 Si+Ti+Nb+Fe+Al atoms | | | | |
| $Si^{4+}$ | 2.11 | 2.20 | 2.03 | 2.21 |
| $Ti^{4+}$ | 3.52 | 3.59 | 3.89 | 3.43 |
| $Al^{3+}$ | 0.10 | | | 0.11 |
| $Fe^{2+}$ | 0.01 | 0.21 | 0.08 | 0.01 |
| $Nb^{5+}$ | 0.26 | | | 0.24 |
| Sum *O* | 3.89 | 3.80 | 3.97 | 3.79 |
| $K^+$ | 0.29 | 0.06 | | 0.01 |
| $Na^+$ | 2.47 | 2.66 | 0.17 | |
| $Sr^{2+}$ | 0.06 | | | |
| $La^{3+}$ | | | 0.96 | 0.91 |
| Sum *A* | 2.83 | 2.72 | 1.13 | 0.92 |
| $OH^-$ | 1.00 | 1.70 | 1.11 | 1.15 |
| $H_2O$ | 4.00 | 4.00 | 4.00 | 4.00 |

[1] Content of $H_2O$ calculated according to sitinakite formula (4 apfu) and OH according to charge-balance requirements.

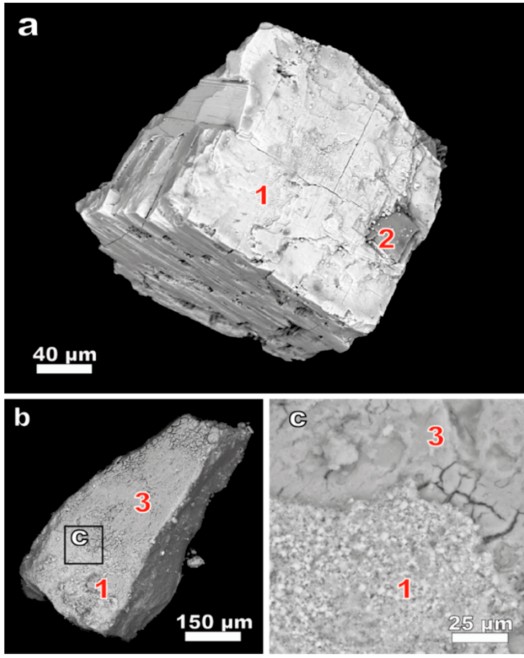

**Figure 2.** Backscattered images of (**a**) La-exchanged natural sitinakite (1) at 200 °C for 24 h with aegirine (2) inclusion; (**b**) synthetic La-exchanged sitinakite at 200 °C for 24 h; (**c**) formation of anatase (3) crusts on the surface of synthetic La-exchanged sitinakite at 200 °C for 24 h.

The size of the individual grains of synthetic sitinakite does not exceed 2 μm which increases the ion-exchange properties but decreases the thermal stability of the material up to 300 °C [42]. The deficiency of Ti in synthetic form might be explained by structural imperfections, which lead to a $TiO_2$ crust formation after ion-exchange experiments. During the ion-exchange experiment, synthetic sitinakite lost 0.13 apfu of Fe and the thin crust of $TiO_2$ appears on the sitinakite grain surface (Figure 2c).

### 3.2. Raman Spectroscopy

The Raman spectra of sitinakite and natural La-exchanged sample are shown in Figure 3. The spectrum of natural sitinakite is similar to that of synthetic sitinakite described in [63]. The assignments of the absorption bands were made by analogy with structurally related titanosilicates [17,63–68] and are given in Table 3.

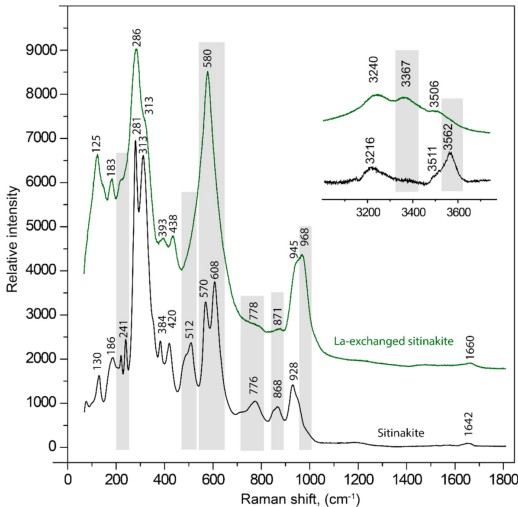

**Figure 3.** Raman spectra of initial sitinakite and La-exchanged natural sitinakite. The most significant differences in the positions or intensity in both spectra are indicated by gray lines.

**Table 3.** Raman shifts in the sitinakite and its La-exchanged form spectra and their assignment.

| Raman Shift, cm$^{-1}$ | | |
|---|---|---|
| **Sitinakite** | **La-Exchanged Sitinakite** | **Assignment** |
| 130 | 125 | *lattice vibrations* |
| 186 | 183 | *lattice vibrations* |
| 241 | | $TiO_6$ |
| 281s | 286s | $TiO_6$ |
| 313s | 313sh | $TiO_6$ |
| 384 | 393 | $SiO_4$ |
| 370 | 378 | $SiO_4$ |
| 420 | 438 | $TiO_6$ |
| 512 | | $TiO_6$ |
| 570s | 580 | $SiO_4$, $TiO_6$ |
| 608s | | $SiO_4$, $TiO_6$ |
| 776 | 778w | $SiO_4$ |
| 868 | 871w | $SiO_4$ |
| 928s | 945 | $SiO_4$ |
| | 965 | $SiO_4$ |
| 1642 | 1660 | $H_2O$ |
| 3216 | 3240 | OH |
| | 3367 | OH |
| 3511 | 3506 | OH |
| 3562 | | $H_2O$ |

sh = shoulder. s = strong intensity. w = weak.

The bands at 928, 868 and 776 cm$^{-1}$ can be attributed to symmetric and asymmetric stretching vibrations of Si–O bonds [63]. The band at 928 cm$^{-1}$ observed in sitinakite shifted and split into two bands at 945 and 968 cm$^{-1}$ in the spectrum of La-exchanged sitinakite, while peaks at 778 and 871 cm$^{-1}$ lost their intensity significantly. Two bands at 570 and 608 cm$^{-1}$ in the sitinakite spectrum are combined into one strong 580 cm$^{-1}$ (Figure 3) band in La-exchanged sitinakite and correspond to the asymmetric bending vibrations of Si–O bonds or overlapping stretching vibrations of Ti–O bonds [17,63]. The same band fusion is caused by the Na $\leftrightarrow$ Cs exchange in sitinakite [63]. The band at 512 cm$^{-1}$ in the sitinakite spectrum disappears in the La-exchanged form and most probably corresponds to the different modes of stretching vibrations of Ti–O bonds in the TiO$_6$ octahedra [17]. The bands in the range 380–450 cm$^{-1}$ are attributed to symmetric bending vibrations of Si–O bonds [67]. The most intensive bands at 281 and 313 cm$^{-1}$ and the low-intensity band at 241 cm$^{-1}$ correspond to the bending vibrations of Ti-O-Si and Ti-O-Ti bonds [63,69,70]. The bands below 200 cm$^{-1}$ belong to translational vibrations.

The weak characteristic bands of the $\nu_2$ bending vibrations of the H$-$O$-$H bonds are observed in the range of 1640–1660 cm$^{-1}$. The bands in the region of 3240–3400 cm$^{-1}$ (Figure 3) correspond to the stretching vibrations in the O$-$H bonds of hydroxyl groups, whereas bands in the range 3506–3570 cm$^{-1}$ correspond to the same vibrations in the H$_2$O molecules [71,72].

During the 3Na $\rightarrow$ La$^{3+}$ ion-exchange, the Raman spectra of sitinakite significantly change and these differences in the position and intensity of bands are highlighted in light-grey color (Figure 3). The most dramatic changes in the range of H$-$O$-$H bending vibrations involve intensity decreasing of the band at 3562 cm$^{-1}$ related to H$_2$O and appearance of the new band at 3367 cm$^{-1}$ related to the O$-$H vibrations in the hydroxyl group. Other changes include the loss of intensity, disappearing bands, shift or fusion bands related to the different modes of Ti$-$O, Si$-$O, Ti-O-Si or Ti-O-Ti vibrations of the titanosilicate framework in the spectrum of La-exchanged sitinakite compared with initial sitinakite.

### 3.3. Powder X-ray Diffraction

Theoretical differences in the XRD powder patterns of the La-exchanged form and sitinakite are shown in Figure 4. The changes in the 8–70° 2θ range include the disappearance of the (002), (200), (211), (220), (404) and (406) reflections (for sitinakite) at 14.74, 22.66, 26.46, 32.30, 56.03 and 66.80° 2θ, respectively. For the La-exchanged sitinakite, additional reflections were observed, which are absent for the initial sitinakite: (111), (202), (401), (043), (440) and (446) at 13.56, 22.00, 33.25, 39.77, 46.47 and 67.40° 2θ, respectively.

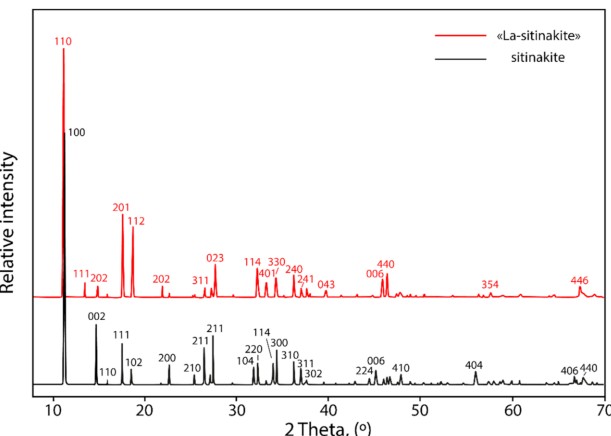

**Figure 4.** Calculated powder XRD pattern of initial sitinakite and La-exchanged form.

The evolution of the PXRD pattern with the increasing exposition in the La(NO$_3$)$_3$ solution is demonstrated in Figure 5. The XRD pattern of the initial sitinakite is in good agreement with theoretical data and agrees well with PDF Card No. 00-050-1689. After 1 h

sorption in the low angle (5–25°) region, the (200) reflection at 22.66° 2θ (3.92 Å) disappears and three new reflections in the high angle region (50–70°) appear: (404) at 56.03° 2θ (1.640 Å), (430) at 58.97° 2θ (1.565 Å) and (406) at 66.80° 2θ (1.399 Å). The characteristic (200) peak in the low-angle region disappears after 4 h sorption, which is related to the formation of La-exchanged sitinakite. For the pattern of the 4h sample, the following changes should be noted: the disappearance of the (211) peak at 26.46° 2θ (3.37 Å) and the appearance of two new peaks in the middle-angle region (043 and 441) at 39.77, 46.47° 2θ as expected from theoretical data. The formation of the La-sitinakite structure is also indicated by the appearance of the (441) peak at 67.40° 2θ (1.390 Å).

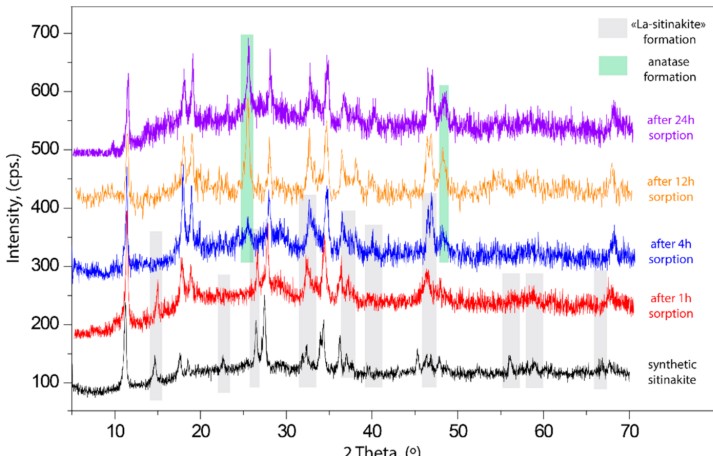

**Figure 5.** Diffraction patterns of synthetic sitinakite and its La-exchanged forms after 1, 4, 12 and 24 h sorption at 200 °C. The most significant differences in the positions or intensity in both patterns are indicated by the gray rectangles.

The crystallization of anatase (TiO$_2$) after 4 h sorption is indicated by green color (Figure 5). The increasing intensity of the anatase reflections at 25.30 and 47.93° 2θ with the increasing sorption time may be related to the partial decomposition of sitinakite owing to the high reactivity of small sitinakite crystallites <2 μm or anatase crystallization from amorphous TiO$_2$ contained in the initial leucoxene concentrate. The peak shape and width of reflections in Figure 5 are indicative of even lower sizes than 2 μm are also important for this material uptake of, e.g., radionuclides (for samples with an extended surface, the mechanism of cation absorption in the pores of the sorbent may predominate).

### 3.4. Single-Crystal X-Ray Diffraction

The crystal structure of natural sitinakite was refined in the $P4_2/mcm$ space group to $R_1 = 0.034$ for 441 independent reflections with $F_o > 4\sigma(F_o)$ ($R_{int} = 0.029$, $R_{sigma} = 0.025$) using the model proposed by Sokolova et al. [73].

The attempt to refine the crystal structure of La-exchanged sitinakite in the same space group resulted in $R_1 = 0.167$ for 438 ($R_{int} = 0.031$, $R_{sigma} = 0.011$) independent reflections. In this space group, the La atoms could not be located. Moreover, the high number of atoms with physically unrealistic displacement parameters and the presence of 287 systematic absence violations ($I > 3\sigma(I)$) were observed. Refinement in the space group $P\bar{4}2m$ provided $R_1 = 0.102$ for 1216 reflections ($R_{int} = 0.031$, $R_{sigma} = 0.020$). This model describes the crystal structure of La-sitinakite better, but more than 500 reflections violating tetragonal symmetry were observed. The careful inspection of additional reflections (Figure 6) allowed the assignment of the *Cmmm* space group as the most possible for La-exchanged sitinakite. The crystal structure was refined using a merohedral twin model (2-fold axis along [110]) with the twin ratio 0.447/0.553 to $R_1 = 0.037$ for 841 a unique observed reflection with $|F_o| \geq 4\sigma$.

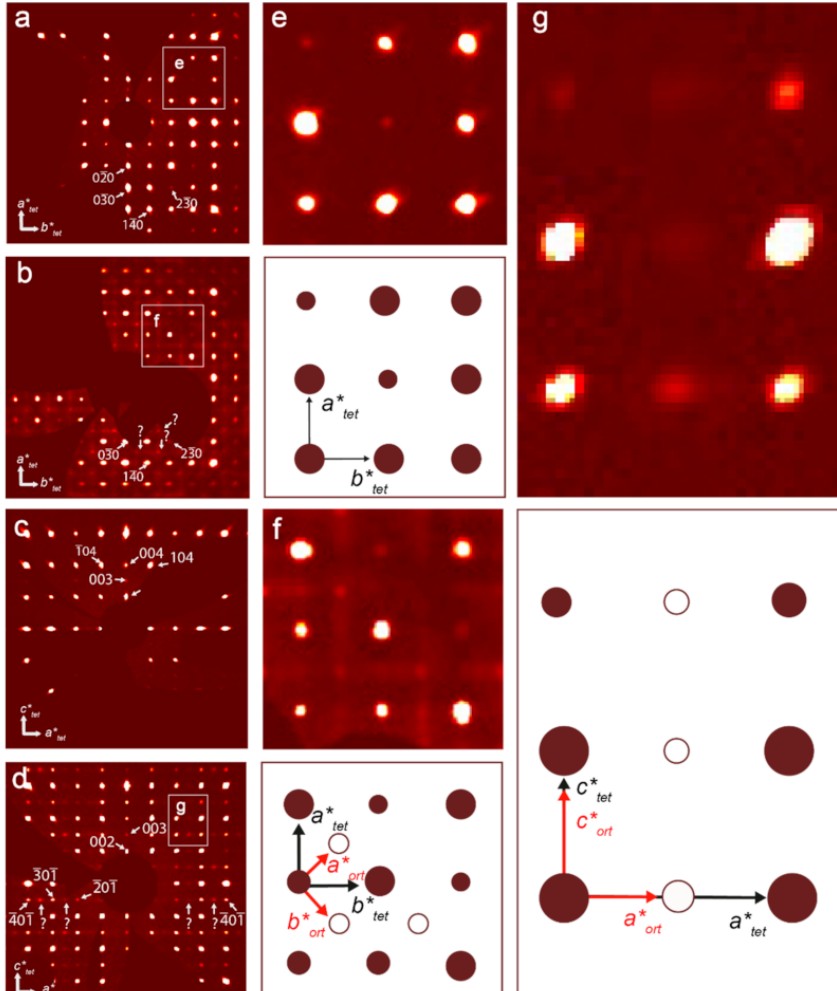

**Figure 6.** Reconstructed sections of reciprocal space obtained for the (*hk*0) and (*h*0*l*) sections for sitinakite (**a**,**c**) and its La-exchanged form (**b**,**d**) and enlarged fragments of these sections (**e–g**). White arrows and numbers indicate reflections and their indices. The examples of additional reflections, which cannot be indexed in the tetragonal cell are indicated by question marks. On the corresponding schemas, large dark red circles and small unfilled circles belong to the tetragonal (*a* = 7.8159, *c* = 12.0167 Å) and orthorhombic (*a* = 11.0339, *b* = 11.0598, *c* = 11.8430 Å) cells, respectively; black and red arrows indicate tetragonal and orthorhombic cell vectors, respectively.

The crystal structure of sitinakite (Figure 7a) and related compounds are based upon cubane-like $[Ti_4O_4]^{8+}$ clusters formed by four edge-sharing $TiO_6$ octahedra [16,74]. The clusters are linked into columns $[Ti_4O_4]^{8+}_\infty$ by sharing vertices along [001] and into the framework by sharing vertices with SiO4 tetrahedra along the [100] and [010] directions in contrast to pharmacosiderite-based compounds, where cubane-like $[Ti_4O_4]^{8+}$ clusters are connected by sharing vertices with $SiO_4$ tetrahedra along the [001], [100] and [010] directions (Figure 7b,c) [73,75]. The heteropolyhedral framework of sitinakite contains a three-dimensional system of crossed channels oriented along the main crystallographic directions of the tetragonal cell. The channel I (Figure 7d) passing along the [001] direction is characterized by an octagonal cross-section with the effective diameter (according to the IUPAC nomenclature, the lengths of the major and minor axes minus the sum of two oxygen ionic radii of 2.7 Å are used for non-isometric channels with an elliptical cross section) [76] equal to $2.78 \times 3.50$ Å$^2$. The channels of the type II are parallel to [100] and [010], whereas the channels of the type III are parallel to [110] (Figure 7e). The channels II and III have hexagonal cross sections and the the effective diameters of $2.16 \times 3.31$ Å$^2$ and

$1.31 \times 3.31$ Å$^2$, respectively. These channels are filled by Na$^+$ and K$^+$ ions, as well as by H$_2$O molecules.

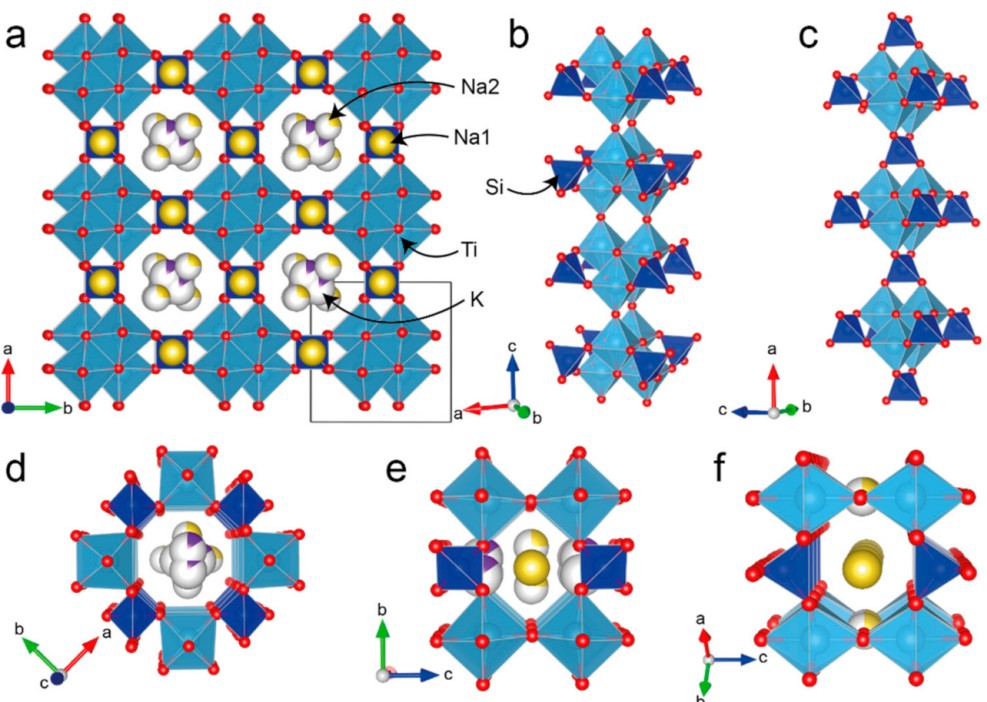

**Figure 7.** The crystal structure of sitinakite projected along the *c* axis (**a**); the [Ti$_4$O$_4$]$^{8+}$$_\infty$ column with adjacent SiO$_4$ tetrahedra in sitinakite (**b**); the connection of [Ti$_4$O$_4$]$^{8+}$ clusters in ivanyukite-K [17] (**c**); the channel I defined by an 8-membered ring (**d**), the 6-membered rings of the channels II (**e**) and III (**f**) with Na and K atoms in the sitinakite structure.

In the sitinakite structure, there is one crystallographically independent Ti site with the Ti-O bonds lengths in the range $1.844-2.127$ Å, with three short bonds ($1.844-1.921$ Å) between Ti$^{4+}$ and O$^{2-}$ and three longer bonds ($2.003-2.127$ Å) between Ti$^{4+}$ and (OH)$^-$. The average <Ti−O> bond distance of 1.970 Å and the polyhedral TiO$_6$ volume of 9.95 Å$^3$ is in agreement with previously reported data for sitinakite with a mean <Ti−O> bond of 1.969 Å and the polyhedral volume of 9.95 Å$^3$ [73]. The SiO$_4$ tetrahedra are characterized by the mean <Si−O> distance of 1.631 Å and the polyhedral volume of 2.23 Å$^3$.

The extra-framework positions include one fully populated Na1 site in octahedral coordination with four bonds of 2.431 Å with O$^{2-}$ framework atoms and two longer bonds of 2.771 Å to extra-framework H$_2$O molecules. Two other extra-framework sites include low-occupied Na2 and K1 sites with occupancies of 0.24 and 0.15, respectively. The Na2 site is 7-coordinated with an average <Na−O> distance of 2.870 Å. As in the original sitinakite structure, the K1 site is slightly shifted from the center of the 8-membered channel I. The K site is 7-coordinated with the mean <K−O> distance of 3.099 Å.

The structural formula of natural sitinakite determined from the structure refinement can be written as Na$_{2.48}$K$_{0.60}$[Ti$_4$O$_2$(O$_{3.08}$[OH$_{0.92}$])$_4$)(SiO$_4$)$_2$]·3.70(H$_2$O).

The crystal structure of La-exchanged sitinakite is shown in Figure 8a–c and differs from that of sitinakite by the presence of extra-framework positions. The framework contains two independent Ti sites (Figure 8d) where the Ti1 site is associated with the intra-channel La1 site. The Ti1 has five short bonds with the Ti1-O distances in the range 1.824–1.988 Å and one longer bond of 2.122 Å. The Ti2 site is accompanied by the La2 and La3 sites and has three short and three long Ti−O distances in the ranges 1.826–1.896 Å and 2.034–2.089 Å, respectively. The polyhedral volumes of the Ti1 and Ti2 sites are almost the same and are equal to 9.82 and 9.81 Å$^3$. The Si1 site has an average <Si−O> distance of 1.631 Å and a polyhedral volume of 2.23 Å.

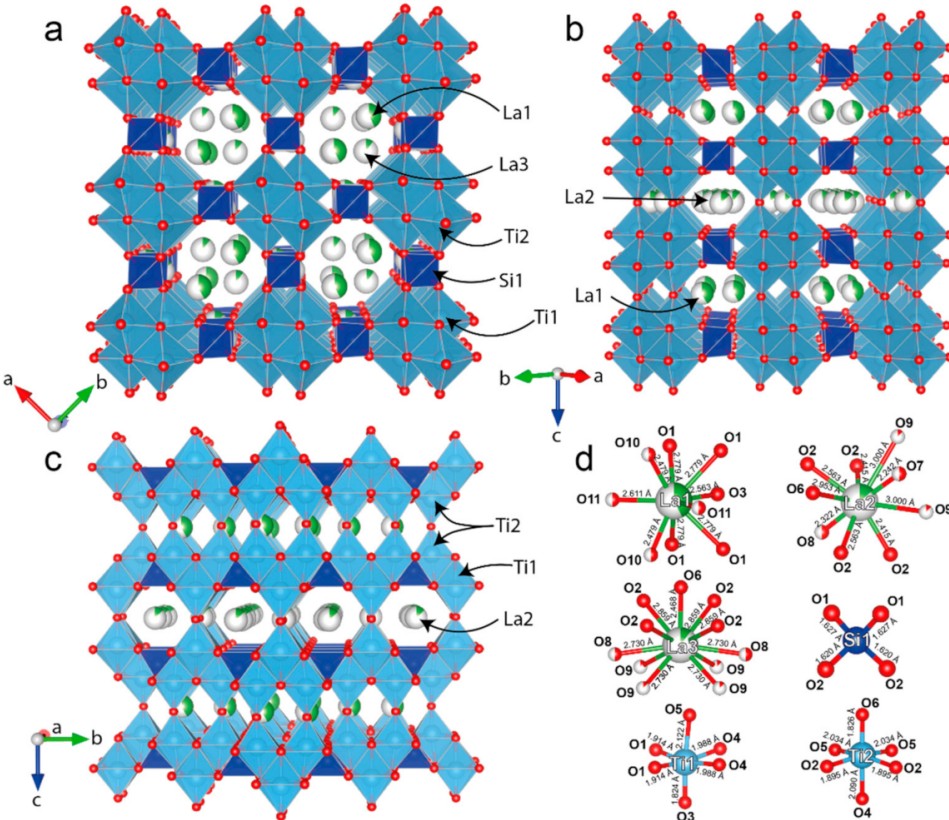

**Figure 8.** The crystal structure of La-exchanged sitinakite projected along *c* axis with channel I (**a**); along [110] direction with channel III (**b**); along *a* axis with channel II (**c**); coordination of La1, La2, La3, Si1, Ti1 and Ti2 atoms (**d**). The occupancy La-sites are indicated by different sectors filled by the green color.

The main feature of the La-exchanged sitinakite is preference occupancy of $La^{3+}$ cations of the 9-coordinated La1 site. There are three extra-framework La sites, La1, La2 and La3, with occupancies equal to 0.42, 0.13, and 0.11, respectively. The La sites are situated at the different *z* levels (Figure 8b,c). Owing to the short La1-La1 distance of 2.838 Å, the La1 sites (Figure 8d) may be half-populated (or less) only. The 9-coordinated La1 site forms five La–O bonds to the framework oxygen atoms related to the pair of $Ti1O_6$ octahedra and the other four La−O bonds to extra-framework low-occupied $H_2O$ molecules. The average <La1−O> distance is 2.693 Å. The La2 site is also 9-coordinated and is split as well with the La2-La2 distance of 1.128 Å. The La3 site is 11-coordinated, with five La3-O bonds to the framework O atoms, two half-populated $H_2O$ molecules and four bonds to low-occupied $H_2O$ molecules.

The structural formula of the La-exchanged sitinakite determined from the structure refinement can be written as $La_{0.79}[Ti_4O_2(O_{2.37}[OH_{1.63}])_4)(SiO_4)_2]\cdot4.32(H_2O)$.

### 3.5. Topological Analysis

The crystal structures of sitinakite and its La-exchanged form are based upon the same 5-nodal net (2,2,3,4,6-coordinated) with the stoichiometry (2-c)(2-c)4(3-c)2(4-c)(6-c)2 (Figure 9). The tiling for this net consists of four types of tiles. Three of them (t-kzd, t-lov, t-cub) are rather small and play the role of gluing large tiles together (Figure 10). The tile $[4^8.6^6.8^2]$ is a new topological type of tile, not previously described in the existing database of tiles (compiled based on an analysis of all zeolites http://iza-structure.org, accessed on 1 August 2021). This tile is a large cavity that can accommodate such large cations as potassium. $Na^+$ cations are located mainly on the edges of this cavity in the center of 6-membered rings.

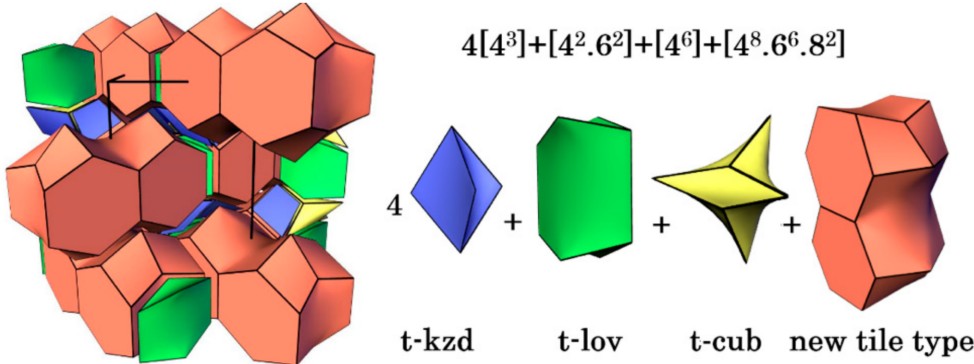

**Figure 9.** Tiling representation of the titanosilicate framework in the crystal structure of sitinakite. The unit cell is outlined.

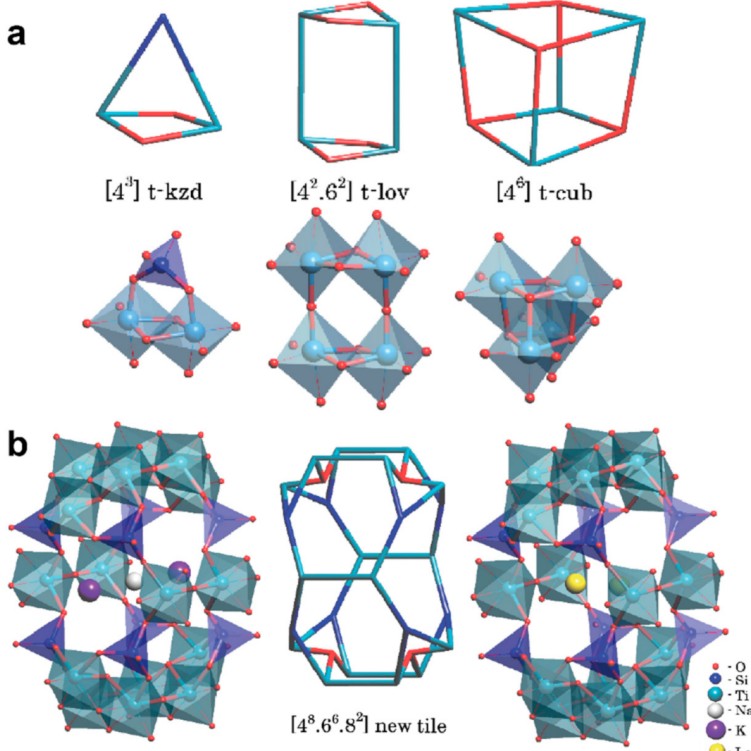

**Figure 10.** The small t-kzd, t-lov, t-cub tiles (**a**) and the large new tile $[4^8.6^6.8^2]$ in the crystal structure of sitinakite. Na, K and La cations are located within the large tile (**b**).

## 4. Discussion

The phase transformation of sitinakite-based materials with transition to 'diagonal' cell ($a$ = ~11.0 Å, $c$ = ~12.0 Å) was reported for H-substituted sitinakite [27], and for Sr-exchanged sitinakite [34]. The same transition was also reported for the dehydrated form of sitinakite [33]. The incorporation of cations with a large effective ionic radius $R \geq 1.40$ Å (like Cs$^+$) does not cause such a transformation [34]. The decreasing symmetry by the scheme $P4_2/mcm \rightarrow Cmmm$ is induced by the incorporation of La into the sitinakite framework and is accompanied by the appearance of ~600 additional reflections (Figure 6).

In all cases, the transformation of the sitinakite unit cell is induced by ion-exchange reactions accompanied by the ion-migration process. According to the Voronoi method, all extra-framework cations (K$^+$, Na$^+$, La$^{3+}$) have 3D possible migration paths (Figure 11). It should be noted that Na$^+$ and La$^{3+}$ cations have more variable paths than K$^+$. The "guest" structure of the extra-framework sites determines the overall symmetry of the material.

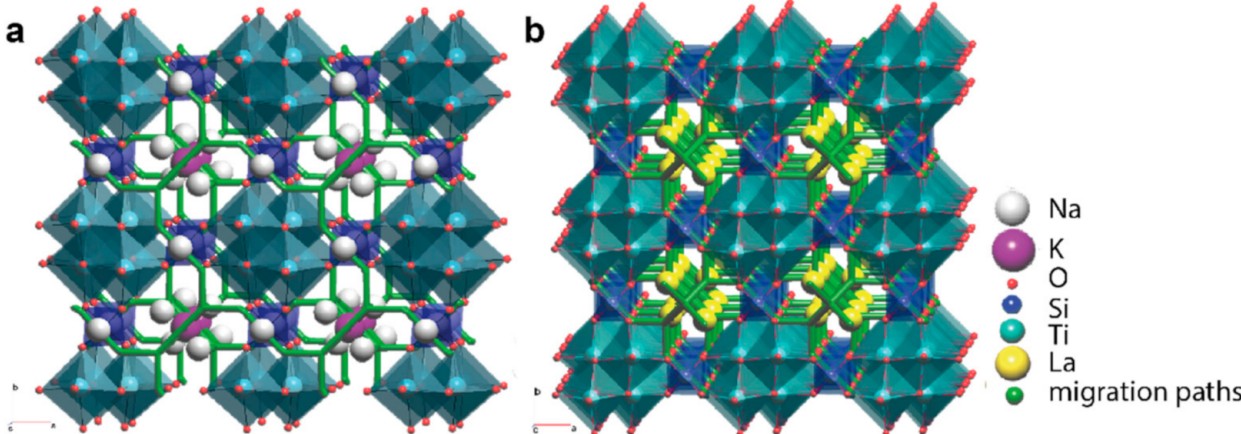

**Figure 11.** Migration paths for Na and K (**a**) for sitinakite and La (**b**) for La-exchanged form.

Ion-exchange processes in the sitinakite-type compounds are accompanied by the rotation of $H_2O$ molecules owing to the $Na^+ \rightarrow Cs^+$ substitution [27] or deprotonation of $OH^-$ groups in the case of the $Na^+ + OH^- \rightarrow Sr^{2+} + O^{2-}$ substitution [42]. Incorporation of $La^{3+}$ into sitinakite structure by the $3Na^+ \leftrightarrow 1La^{3+} + 2\square$ substitution leads to the rearrangement of the hydrogen bonding system, which is reflected in the changes in the range of the $H{-}O{-}H$ stretching vibrations observed in Raman spectra. According to the bond-valence sum (BVS) analysis (Table 4), in the initial sitinakite, there is only one partially protonated O2 site (BVS = 1.63), whereas the O1 and O3 atoms has BVSs close to 2. In La-exchanged sitinakite, only O5 atom is associated with an $OH^-$ group with the BVS of 1.56. The symmetry breaking according to the group-subgroup $P4_2/mcm \rightarrow Cmmm$ transition results in the splitting of the O2 site in sitinakite into the O5 and O4 atoms in La-exchanged sitinakite. Most probably, additional band at 3367 cm$^{-1}$ in the Raman spectrum of La-exchanged sitinakite is related to the $O{-}H$ vibrations in the O5-H hydroxyl group.

**Table 4.** Bond-valence sum (in valence units, v.u.) for the framework-related atoms in sitinakite and La-exchanged sitinakite structure.

| | La-Exchanged Sitinakite | | | | Sitinakite | | |
|---|---|---|---|---|---|---|---|
| site | Ti1 | Ti2 | Si1 | sum/sum * | Ti1 | Si1 | sum/sum * |
| O1 | 0.765↓×2 | | 0.992↓×2 | 1.76/1.96 | 0.751 | 0.987↓×4 | 1.74/1.98 |
| O2 | | 0.803↓×2 | 1.014↓×2 | 1.82/1.93 | 0.602↓×2→×2 0.430 | | 1.63/1.63 |
| O3 | 0.976→×2 | | | 1.95/2.03 | 0.926→×2 | | 1.85/1.90 |
| O4 | 0.627↓×2→×3 | | | 1.88/1.92 | | | |
| O5 | 0.436 | 0.553↓×2→×2 | | 1.54/1.56 | | | |
| O6 | | 0.971→×2 | | 1.94/1.96 | | | |
| Sum | 4.20 | 4.16 | 4.01 | | 4.24 | 3.95 | |

\* Considering bonds with extra-framework atoms. Calculated using the bond valence parameters from the [77].

It should be noted the slight excess of extra-framework cations ($Na^+ + La^{3+}$) over their amount allowed by only the ion-exchanged mechanism for synthetic sitinakite. Initially, synthetic sitinakite contains 2.72 apfu of extra-framework cations with a total charge of 2.72. The La-exchanged form of synthetic sitinakite contains 1.13 apfu extra-framework cations with a total charge of 3.05. The proposed additional mechanism for sorption includes substitution $\square + 3OH^- \leftrightarrow 1La^{3+} + 3O^{2-}$. The same additional sorption mechanism was observed for divalent cations in [1].

The ordering of La in sitinakite structure is the main reason for the unit-cell transformation from tetragonal $a = 7.8159$, $c = 12.0167$ to orthorhombic $a = 11.0339$, $b = 11.059$, $c = 11.8430$ Å. The fully populated Na1 site in sitinakite is substituted by the La2 site in the La-exchanged form with the occupancy of 0.13. The $\frac{1}{4}$-occupied Na2 site is substituted by

the La1 and La3 sites at different *z* levels with the occupancies of 0.42 and 0.11, respectively (Figure 12a,b). The La1 site concentrates ~2/3 of the total amount of La in the structure. Incorporation of $La^{3+}$ into the La1 site increases the O4−O4 distance up to 4.062 Å and the O3−O3 distance up to 7.96 Å. At the same time, the corresponding distances in the void associated with the low-occupied La3 site decrease to 3.982 and 7.72 Å, respectively (Figure 12c). The uneven La distribution causes the distortion of a rhombus formed by adjacent $TiO_6$ octahedra. The polyhedral tension of the La1 site results in the decreasing of the O3−O3 distance to 3.10 Å and the distance between the two nearest Ti1−Ti1 sites to 3.02 Å (Figure 12d). For the neighboring cavity with the La3 site, the similar O6−O6 and Ti2−Ti2 distances are 3.32 and 3.18 Å, respectively (Figure 12e). The dominant occupancy of the La1 site by La also causes the redistribution of $OH^−$ groups in the structure, whereas the O5 site is associated with the low-occupied La3 site.

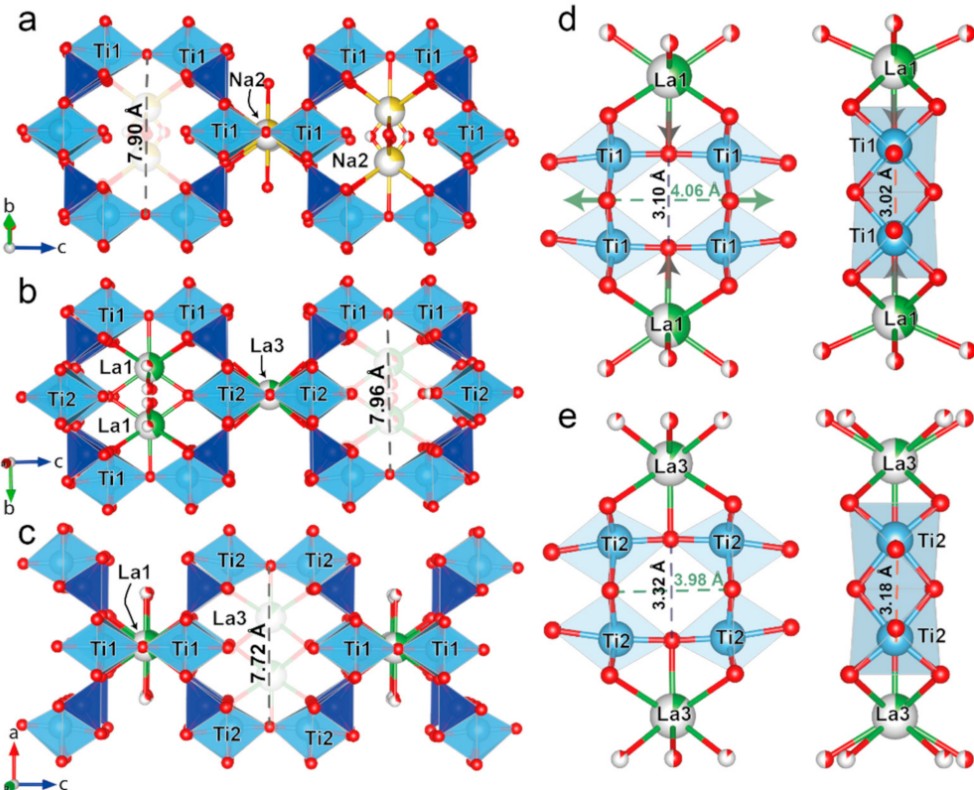

**Figure 12.** Largest cavity corresponding to the tile $[4^8.6^6.8^2]$: in sitinakite with the Na2 site (**a**); in La-exchanged sitinakite with the La1 site (**b**); in La-exchanged sitinakite with La3 site (**c**); the hinge-like deformation in the $Ti1O_6$ octahedra associated with the La1 site (**d**); and associated with the La3 site (**e**). Green and grey arrows represent distortions caused by the La incorporation.

The flexibility of compounds based upon cubane-like titanate $[Ti_4O_4]^{8+}$ clusters is related to the hinge-like deformations [78] induced by an ion-exchange [79] or ion-migration processes in ivanyukite-type compounds [17]. In contrast to ivanyukite, the sitinakite structure contains additional hinges oriented along the *c* axis with neighboring $[Ti_4O_4]^{8+}$ clusters sharing vertices to form the $[Ti_4O_4]^{8+}_{\infty}$ columns. The additional hinges (the O6 and O3 sites in La-exchanged sitinakite and the O5 site in sitinakite) make the expected transformations more variable.

Along with additional reflections induced by the symmetry breaking, some diffuse reflections have been observed (Figure 6f). However, these reflections are outside the resolution limits and could not be considered. The modulated arrangement of $H_2O$ sites (Figure 13) along [010] is probably connected with the character of the low-occupied sites and produces diffuse reflections.

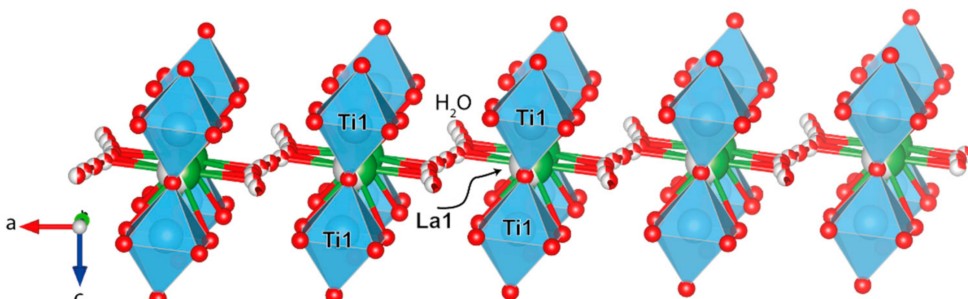

**Figure 13.** The modulated character of low-occupied $H_2O$ sites along [010] direction associated with La1 sites.

## 5. Conclusions

For the first time, sitinakite and their synthetic analogs were applied for the incorporation of $La^{3+}$ cations. Incorporation of $La^{3+}$ into sitinakite structure by the $3Na^+ \leftrightarrow 1La^{3+} + 2\square$ substitution leads to the rearrangement of the hydrogen bonding system, which is reflected in the changes in the range of the H−O−H stretching vibrations observed in Raman spectra. Incorporation of $La^{3+}$ substitute all $Na^+$ and $K^+$ in the initial structure and may involve additional mechanism for La sorption: $\square + 3OH^- \leftrightarrow 1La^{3+} + 3O^{2-}$.

Ordering of $La^{3+}$ in the structure sitinakite accompanied by mechanisms of the crystal chemical adaptation reflects with Na to La ion-exchange in sitinakite involve variable occupancies of the La sites at different *z* levels. The different occupancies of the La sites lead to the redistribution of H-atoms in sitinakite, on the one hand, and the distortion of $TiO_6$ octahedra, on the other hand. The main mechanism of the structural adaptation in sitinakite involves hinge-like deformations inside the cubane-like $[Ti_4O_4]^{8+}$ clusters caused by the ordered occupancy of La in the cavities leading to the symmetry decrease from $P4_2/mcm$ in sitinakite to *Cmmm* for its La-exchanged form. The breaking of the symmetry accompanied by the appearance of more than 200 additional reflections was determined from SC XRD data.

Increasing the time interactions of synthetic sitinakite with $La(NO_3)_3$ solution lead to the formation of the $TiO_2$ phase.

**Supplementary Materials:** The following supporting information can be downloaded at: https://www.mdpi.com/article/10.3390/min12020248/s1. Table S1: Atomic coordinates for sitinakite, Table S2: Atomic coordinates for La-exchanged sitinakite, Table S3: Selected interatomic distances for sitinakite, Table S4: Selected interatomic distances for La-exchanged sitinakite, Table S5: Anisotropic displacement parameters for sitinakite, Table S6: Anisotropic displacement parameters for La-exchanged sitinakite.

**Author Contributions:** Conceptualization, T.L.P.; methodology, G.O.K.; software, N.A.K.; validation, V.N.Y.; investigation, I.A.P., A.V.B. and V.N.B.; writing—original draft preparation, T.L.P., G.O.K. and N.A.K.; writing—review and editing, T.L.P., I.A.P, S.V.K. and A.I.N.; visualization, N.A.K.; supervision, S.V.K.; funding acquisition, A.I.N. and T.L.P. All authors have read and agreed to the published version of the manuscript.

**Funding:** This research was funded by the Russian Foundation for Basic Research, grant 18-29-12039 and Russian Science Foundation, project no. 21-77-10103, State tasks, AAAA-A17-117020110035-5 and AAAA-A19-119111190038-5.

**Institutional Review Board Statement:** Not applicable.

**Informed Consent Statement:** Not applicable.

**Data Availability Statement:** Not applicable.

**Acknowledgments:** Authors are grateful to the X-ray Diffraction Centre, Geo Environmental Centre "Geomodel" of Saint-Petersburg State University for experimental studies.

**Conflicts of Interest:** The authors declare no conflict of interest.

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
