# Peer review of "Ion-Exchange-Induced Transformation and Mechanism of Cooperative Crystal Chemical Adaptation in Sitinakite: Theoretical and Experimental Study"

_minerals, doi:10.3390/min12020248_

Round 1
Reviewer 1 Report
The authors discussed the theoretical and experimental study of ion-exchange-induced transformation and mechanism of cooperative crystal chemical adaptation in sitinakite. The manuscript is well written and is very interesting to the community. I recommend to accept the manuscript for publication.
Author Response
There is no comments from 1st reviewer
Reviewer 2 Report
The manuscript by Panikorovskii et al. reports the results of a comprehensive (combining theoretical and empirical methods) study drawing light on the mechanism of cooperative crystal chemical adaptation caused by the incorporation of La3+ ions in the structure of the microporous hydrous silicotitanate - sitinakite (both mineral and synthetic forms). This material and its modifications known as IONSIV IE-911, CST, PCST, IONSIV R9120, etc. are well known efficient exchangers for radionuclides. Their performance has already been reported after their utilization for nuclear waste effluent treatment processes (Fukushima Daiichi plant after the events from March 11, 2011). Thus, any new information about the crystal chemical specifics and behaviour of the studied material will be relevant both in practical and scientific terms. The manuscript can be accepted for publication after minor revisions and addition of certain comments.
More important comments and suggestions:
Rows 371 and 396 provide information for the structural formula of sitinakite and La-exchanged sitinakite determined from the structure refinement of presumably natural sample and its ion-exchanged form (although this is not duly specified). Table 2 provides information for the chemical compositions of the same materials. The obvious compositional discrepancies should be addressed as follows:
1) All presented in Table 2 compositions reveal slight excess of silicon and deficiency of titanium that has not been observed in the structural formulas which do not include also the octahedral Nb5+, Al3+, etc. (rows 371-372, 395-396). Already the first researchers of the closely structurally related material (the pharmacosiderite-type structure titanosilicate GTS-1) noticed a slight excess of silicon over titanium in the framework.
2) The excess of silicon could be due to the presence of amorphous silica in the run products of the as-synthesized and its La-exchanged forms (indeed an amorphous halo can be observed in the PXRD patterns presented in Fig. 5). Its presence there has not been commented but such source of extra silica is hardly expected for the unprocessed mineral form.
3) On its side, the deficiency of titanium (taking into account also the increasing amount of anatase registered by the PXRD for the synthetic form) might be explained with structural imperfections affecting the channels’ construction by depletion/absence of this element from its expected structural positions.
4) The charge disbalance (most noticeable for the as-synthesized sitinakite - data from Table 2) is another indication for possible structural imperfections (There is also the lack of Nb5+ contribution to the positive charge).
5) The slight excess of extra-framework cations (Na+ and La3+) over their amount allowed by only the ion-exchanged mechanism probably indicate additional mechanism for sorption of lanthanum controlled by the titanium shortage and increasing the adsorption capacity and rapid uptake of radionuclides for the studied materials.
6) Size of crystallites (only the upper limit of 2 µm has been announced although peak shapes and width of reflections in Fig. 5 are indicative of even lower sizes) are also important for this material uptake of e.g. radionuclides.
One of the “must” requirements of the journal is the presence of a conclusion part in the submitted manuscripts. In this attitude the rows 481-483 of the last paragraph should be either removed or moved into more appropriate sections – Introduction and/or Methodological part. The rest of it should be reworked accordingly.
Additional minor comments refer to:
Abstract, rows 23-24. Seems like parts of two sentences have been combined in only one.
Row 34. Why the word “Arctic” is among the key words? Is it only a tribute to the corresponding author affiliation or it gives something more? ;)
Rows 224-226. It is not clear to me the use of the word “instead” in this sentence.
Rows 228-229. This sentence should precede and appear in the beginning of the first paragraph in part 3 Results.
Figure 4. Designation of reflection indices (better d-spacings or 2 Theta values) directly in the graph would facilitate the understanding of text above. However this means to introduce the two space groups - P42/mcm and Cmmm already here and not in section 3.4. Single-Crystal X-Ray Diffraction…..
Rows 173, 283 – why do you use the term “theoretical”? Actually these are calculated PXRD patterns simulated/obtained on the basis of single crystal-structure data by means of the VESTA 3 program………….
Row 323. Were instead of was.
Row 331. Their instead of its.
Rows 360-362. Not clear.
Rows 404, 409, 440. La3+ is almost equal in size with Na+ (seen also from rows 194-195). Consider this when depicting it in the legends of Figs. 10 and 11. See also row 404.
Row 433. Have instead of has.
Fig. 8, Rows 386-…, 449-…. A brief comment on the preference of lanthanum for its positioning in the structure will be very relevant.
There are places in the manuscript where it is not clearly stated whether the data presented concern the natural or the synthetic material under study, e.g. 312-316; 371, 395, Table 2, etc. Please specify duly this throughout the whole manuscript.
Author Response
2nd reviewer
The manuscript by Panikorovskii et al. reports the results of a comprehensive (combining theoretical and empirical methods) study drawing light on the mechanism of cooperative crystal chemical adaptation caused by the incorporation of La3+ ions in the structure of the microporous hydrous silicotitanate - sitinakite (both mineral and synthetic forms). This material and its modifications known as IONSIV IE-911, CST, PCST, IONSIV R9120, etc. are well known efficient exchangers for radionuclides. Their performance has already been reported after their utilization for nuclear waste effluent treatment processes (Fukushima Daiichi plant after the events from March 11, 2011). Thus, any new information about the crystal chemical specifics and behaviour of the studied material will be relevant both in practical and scientific terms. The manuscript can be accepted for publication after minor revisions and addition of certain comments.
More important comments and suggestions:
Rows 371 and 396 provide information for the structural formula of sitinakite and La-exchanged sitinakite determined from the structure refinement of presumably natural sample and its ion-exchanged form (although this is not duly specified). Table 2 provides information for the chemical compositions of the same materials. The obvious compositional discrepancies should be addressed as follows:
1) All presented in Table 2 compositions reveal slight excess of silicon and deficiency of titanium that has not been observed in the structural formulas which do not include also the octahedral Nb5+, Al3+, etc. (rows 371-372, 395-396). Already the first researchers of the closely structurally related material (the pharmacosiderite-type structure titanosilicate GTS-1) noticed a slight excess of silicon over titanium in the framework.
Response:
We are very grateful to reviewers for the constructive comments. We tried to revise our manuscript as best as we can. Authors made corrections as suggested reviewer. The admixture of Nb5+, Al3+ is not exceed 5% of occupancy of Ti1 site and not refined. It should be noted that
Lines 258-259 changed accordingly:
The slight excess of Si (2.03-2.21 apfu) also observed for pharmacosiderite-type structure titanosilicate GTS-1 [64].
2) The excess of silicon could be due to the presence of amorphous silica in the run products of the as-synthesized and its La-exchanged forms (indeed an amorphous halo can be observed in the PXRD patterns presented in Fig. 5). Its presence there has not been commented but such source of extra silica is hardly expected for the unprocessed mineral form.
Response:
Lines 260-262 changed accordingly:
The most significant Si excess observed for the synthetic form and could be due to the presence of amorphous silica. For the natural sample slight Si excess may be connected with possible substitution Si4+ + Ti4+ Al3+ + Nb5+.
3) On its side, the deficiency of titanium (taking into account also the increasing amount of anatase registered by the PXRD for the synthetic form) might be explained with structural imperfections affecting the channels’ construction by depletion/absence of this element from its expected structural positions.
Response:
Lines 265-266 changed accordingly:
The deficiency of Ti in synthetic form might be explained with structural imperfections, which lead to a TiO2 crust formation after ion-exchange experiments.
4) The charge disbalance (most noticeable for the as-synthesized sitinakite - data from Table 2) is another indication for possible structural imperfections (There is also the lack of Nb5+ contribution to the positive charge).
Response:
The charge balance in sitinakite determined by O/OH ratio (this data already present in table 2)
5) The slight excess of extra-framework cations (Na+ and La3+) over their amount allowed by only the ion-exchanged mechanism probably indicate additional mechanism for sorption of lanthanum controlled by the titanium shortage and increasing the adsorption capacity and rapid uptake of radionuclides for the studied materials.
Response:
We add lines 480-486 in the Discussion section:
It should be noted the slight excess of extra-framework cations (Na+ + La3+) over their amount allowed by only the ion-exchanged mechanism for synthetic sitinakite. Initially synthetic sitinakite contain 2.72 apfu of extra-framework cations with total charge of 2.72. La-exchanged form of synthetic sitinakite contain 1.13 apfu extra-framework cations with total charge of 3.05. The proposed additional mechanism for sorption includes substitution □ + 3OH– ↔ 1La3+ + 3O2–. The same additional sorption mechanism observed for divalent cations in [1]
6) Size of crystallites (only the upper limit of 2 µm has been announced although peak shapes and width of reflections in Fig. 5 are indicative of even lower sizes) are also important for this material uptake of e.g. radionuclides.
Response:
Lines 342-345 changed accordingly:
The peak shape and width of reflections in Fig. 5 are indicative of even lower sizes than 2 µm are also important for this material uptake of e.g. radionuclides (for samples with an extended surface, the mechanism of cation absorption in the pores of the sorbent may predominates).
One of the “must” requirements of the journal is the presence of a conclusion part in the submitted manuscripts. In this attitude the rows 481-483 of the last paragraph should be either removed or moved into more appropriate sections – Introduction and/or Methodological part. The rest of it should be reworked accordingly.
Response:
We add conclusion section (Lines 531-550):
For the first time sitinakite and their synthetic analogs were applied for incorporation of La3+ cations. Incorporation of La3+ into sitinakite structure by the 3Na+ ↔ 1La3+ + 2□ substitution leads to the rearrangement of hydrogen bonding system, which is reflected in the changes in the range of the H−O−H stretching vibrations observed in Raman spectra. Incorporation of La3+ substitute all Na+ and K+ in the initial structure and may involve additional mechanism for La sorption: □ + 3OH– ↔ 1La3+ + 3O2–.
Ordering of La3+ in the structure sitinakite accompanied by mechanisms of the crystal chemical adaptation reflects with Na to La ion-exchange in sitinakite involve variable occupancies of the La sites at different z levels. The different occupancies of the La sites leads to the redistribution of H-atoms in sitinakite, on the one hand, and the distortion of TiO6 octahedra, on the other hand. The main mechanism of the structural adaptation in sitinakite involves hinge-like deformations inside the cubane-like [Ti4O4]8+ clusters caused by the ordered occupancy of La in the cavities leading to the symmetry decrease from P42/mcm in sitinakite to Cmmm for its La-exchanged form. The breaking of the symmetry accompanied by appearing of more 200 additional reflections determined from SC XRD data.
Increasing of the time interactions of synthetic sitinakite with La(NO3)3 solution lead to the formation of TiO2 phase.
Additional minor comments refer to:
Abstract, rows 23-24. Seems like parts of two sentences have been combined in only one.
Row 34. Why the word “Arctic” is among the key words? Is it only a tribute to the corresponding author affiliation or it gives something more? ;)
Rows 224-226. It is not clear to me the use of the word “instead” in this sentence.
Rows 228-229. This sentence should precede and appear in the beginning of the first paragraph in part 3 Results.
Figure 4. Designation of reflection indices (better d-spacings or 2 Theta values) directly in the graph would facilitate the understanding of text above. However this means to introduce the two space groups - P42/mcm and Cmmm already here and not in section 3.4. Single-Crystal X-Ray Diffraction…..
Rows 173, 283 – why do you use the term “theoretical”? Actually these are calculated PXRD patterns simulated/obtained on the basis of single crystal-structure data by means of the VESTA 3 program………….
Row 323. Were instead of was.
Row 331. Their instead of its.
Rows 360-362. Not clear.
Rows 404, 409, 440. La3+ is almost equal in size with Na+ (seen also from rows 194-195). Consider this when depicting it in the legends of Figs. 10 and 11. See also row 404.
Row 433. Have instead of has.
Fig. 8, Rows 386-…, 449-…. A brief comment on the preference of lanthanum for its positioning in the structure will be very relevant.
There are places in the manuscript where it is not clearly stated whether the data presented concern the natural or the synthetic material under study, e.g. 312-316; 371, 395, Table 2, etc. Please specify duly this throughout the whole manuscript.
Response:
Sitinakite was found in the Khibiny massif in the Russian Arctic, so we use keyword “Arctic” in order to highlight its geographic.
All minor corrections fixed as suggested reviewer.

Reviewer 3 Report
The manuscript "Ion-exchange-induced transformation and mechanism of cooperative crystal chemical adaptation in sitinakite: theoretical and
experimental study" fits within the scope of the Journal. It is well written, and with a few minor modifications and suggestions, I would assume it would be close to a publishable version.
1) Please consider reducing the reference count. Just the introduction alone cites 38 articles. Even though there is no limit set by the Journal, I am not convinced you would need almost 40 articles to bring up the state of the art of your manuscript.
2) Still on the introduction, bear in mind that this is not a literature review. Be more concise and explain in greater detail the research gap you have been trying to explore with your paper.
3) Can you detail the methodology used in line 128?
4) Please avoid excessive self-citation to detail the state of the art of the research scope, as seen in lines 482-483.
Results and Discussion are fine.
Author Response
3rd reviewer
The manuscript "Ion-exchange-induced transformation and mechanism of cooperative crystal chemical adaptation in sitinakite: theoretical and
experimental study" fits within the scope of the Journal. It is well written, and with a few minor modifications and suggestions, I would assume it would be close to a publishable version.
- Please consider reducing the reference count. Just the introduction alone cites 38 articles. Even though there is no limit set by the Journal, I am not convinced you would need almost 40 articles to bring up the state of the art of your manuscript.
Response:
We are very grateful to reviewers for the constructive comments. The Sitinakite-lake compounds are one of the most famous ion-exchangers and all 40 citations are mandatory for introduction.
- Still on the introduction, bear in mind that this is not a literature review. Be more concise and explain in greater detail the research gap you have been trying to explore with your paper.
Response:
We agree. The lines 96-105 changed accordingly:
In our recent works, we demonstrated different mechanisms of structure adaptation induced by the cation ordering causing symmetry breaking in vesuvianite [38,39], garnet- [40] and eudialyte- [41] group minerals. Mechanisms of incorporation of mono and divalent cations are intensively reported during last years [42,43], however there is a lack information about incorporation of trivalent cations in sitinakite structure.
Owing to endemic status of the mineral the single-crystal X-ray diffraction (SC XRD) studies of ion-exchange mechanisms in sitinakite is quite difficult for studies. Additionally, size of synthetic counterparts also often is not appropriate for SC XRD. Herein we report the incorporation of La3+ into natural sitinakite at 200 °C and the structural adaptation mechanism based on the SC XRD data.
- Can you detail the methodology used in line 128?
We add detailed description (Lines 135-163)
A hydrated precipitate obtained by the original fluorine-ammonium method for leucoxene concentrate (Yarega deposit, Komi Republic, Russia) processing [48] was used as a precursor for the synthesis of titanosilicates. The enrichment of the leucoxene concentrate was carried out in a tubular furnace equipped with a gas outlet system that allows volatile products (gaseous ammonia, water, volatile fluoride compounds) to be pumped out into a vessel with water. Concentrate was mixed with ammonium hydrofluoride in a glassy carbon crucible without additional abrasion and placed in the center of the furnace, where two step heating was carried out: at 220 (heating rate was 10 °C/min) and 300 °C (heating rate was 2 °C/min). Sample was aged for 30 min at each temperature. After fluorination, water leaching (deionized water heated to 70 °C with a volume of 50 cm3, рН = 5.20±0.10) was carried out, which made it possible to transfer undecomposed fluoride complexes into solution via hydrolysis. The remainder of the titanium enriched concentrate after leaching was separated from the mother solution by filtration and dried at 103 °C. The fluorinated product is a high-titanium concentrate containing more than 85% TiO2 and represented by a mixture of rutile and anatase. The fluorinated product can be processed into pigmented titanium dioxide or into metallic titanium.
The waste product on fluorination, the mother solution, is a multi-component system of dissolved fluorammonium salts. Addition of ammonia (6 % NH4OH) to the mother solution leads to gradual aggregation of colloidal particles and formation of a hydrated precipitate. The chemical composition of the hydrated precipitate was determined using X-ray fluorescence analysis: 50.5% - TiO2; 45.5% - SiO2; 2.9% - Fe2O3; 0.4% - Al2O3; 0.2% - CaO; 0.2% - K2O; 0.2% - NbO and ZrO2. The dried hydrated precipitate in the amount of 0.5 g was treated with 37 mL of 1 M NaOH solution and dispersed for 20 min with a magnetic stirrer. The final mole ratio of Na2O:TiO2:SiO2:H2O in the resulting alkali titanium-silicon mixture was 6:1:1.2:657.7. The obtained mixture was transferred to Teflon-lined autoclave (45 ml, filling degree 80%) and kept at 250°C for 12 hours (pressure in the autoclave can be estimated as ~80 atm). After cooling down to room temperature (20◦С), the product was collected by centrifugation and washed with distilled water (450 ml) until pH 5.6–6. The sample was dried at 103 ◦C for 4 h.
- Please avoid excessive self-citation to detail the state of the art of the research scope, as seen in lines 482-483.
Response:
We exclude this citations from discussion part. They are appropriate in the introduction section.
Results and Discussion are fine.
